# Study on Wear Mechanism of Helical Gear by Three-Body Abrasive Based on Impact Load

**DOI:** 10.3390/ma15124135

**Published:** 2022-06-10

**Authors:** Wei Yuan, Haotian Wang, Qianjian Guo, Wenhua Wang, Yuqi Zhu, Jie Yu, Xianhai Yang

**Affiliations:** School of Mechanical Engineering, Shandong University of Technology, Zibo 255000, China; wyuan16@sdut.edu.cn (W.Y.); wanghaotian3023@126.com (H.W.); 17853317785@163.com (W.W.); zhuyuqi1730@163.com (Y.Z.); yujiezb7@sdut.edu.cn (J.Y.); yxh@sdut.edu.cn (X.Y.)

**Keywords:** gear, initial hard particle, impact, oil analysis, vibration analysis

## Abstract

This study aimed to explore the wear characteristics and evolution mechanisms of large-scale wind power gears under the impact load of particles of the three-body abrasive Al_2_O_3_ (0.2 mg/mL) from four aspects: oil analysis, vibration analysis, amount of gear wear, and tooth-surface-wear profile analysis. A magnetic powder brake was used to simulate the actual working conditions. Combined with the abrasive particle monitoring and the morphology analysis of the tooth-surface-wear scar, by setting quantitative hard particles in the lubricating oil, the gears are mainly operated in the abrasive wear state, and wear monitoring and wear degree analysis are carried out for the whole life cycle of the gears. Oil samples were observed and qualitatively analyzed using a particle counter, a single ferrograph, a metallographic microscope, and a scanning electron microscope. The experiments demonstrate that the initial hard particles have a greater impact in the early wear stage of the gears (<20 h), and abrasive particle concentration increases by 30%. This means that Al_2_O_3_ particles accelerate the gear wear during the running-in period. The loading method of the impact load on the oblique gear exacerbates the abrasion particle wear and expands the stress concentration, which reduces the surface of large milling particles on the surface, and reduces the width of the tooth (the part above the pitch line is severely worn), which causes the gear to break into failure. The research provides help for analyzing the mechanism of abrasive wear of gears and predicting wear life.

## 1. Introduction

The running of large wind power equipment relies on the operation of its internal gears. The operation of gears is affected by fluctuating loads and harsh environments. Hard particles inevitably mix in during the early work and, thus, lead to the failure of the gears. The main reasons for gear failure are wear and fatigue cracking [1]. The failure of gears makes a lot of noise, increases the sense of frustration, increases the mutual friction rate between the gears, increases the temperature, as well as breaking the gear teeth under severe impact loads [2]. Therefore, finding effective ways to improve the quality of gears is important and urgent in the gear manufacturing industry [3]. Since current technologies only minimize the amount of gear wear, new methods need to be studied to completely avoid gear wear.

Currently, more researchers are focusing on effective ways to monitor the wear state of gears, such as oil monitoring and vibration signal monitoring [4,5,6]. Peng et al. [7] develop an automatic wear and abrasive particle detection and analysis system, which provides development conditions for the application of online machine condition monitoring. Based on oil monitoring technology, Cao et al. [8] improve the online wear prediction model and the wear prediction accuracy. Ashwani et al. [9] develop a model based on ferrography analysis technology to predict the degree of wear. Sun et al. [10] use dual-tree complex wavelet transform (DTCWT) to obtain the features of multi-scale signals, which improves the recognition accuracy of fault characteristic signals. Moumene et al. [11] propose an improved gear and bearing fault diagnosis method, using an optimized wavelet packet transform (OWPT) for signal denoising and fault feature extraction. Barbour et al. [12] propose a hybrid method based on fully ensemble empirical mode decomposition, with adaptive noise, optimized wavelet multiresolution analysis, and Hilbert transforms, which has practical significance for detecting potential mechanical faults. Lior et al. [13] conduct experimental simulations on healthy gears and faulty gears through studying the influence of surface roughness and the influence of different working conditions and gear faults on gear vibration. The information processing technology improved by Sahoo et al. [14] is more suitable for fault identification and state detection of gears. Li et al. [15] propose a feature extraction fault diagnosis method, based on the multi-center frequency and vibration signal spectrum. Therefore, online detection technology occupies a dominant position. The online abrasive detection technology has the characteristics of short waiting times and convenient operation [16]. However, this technology also has some shortcomings, such as a small detection range of abrasive particles, and imperfect automatic recognition of abrasive images.

As shown in previous works [17,18], the method for analyzing gear wear is relatively simple. At the same time, complex and changeable real-world environmental factors have not yet been simulated. Combined with the wear particle monitoring and the microscopic analysis of the tooth-surface-wear scar morphology, by setting quantitative hard particles in the lubricating oil, the wear monitoring and wear degree analysis of the gear life cycle are carried out. In this paper, we built a test bench, set the initial condition control variables, and used vibration analysis technology and oil analysis technology to monitor the wear state of the gearbox under the impact load of Al_2_O_3_ hard particles. The experiment used a scanning electron microscope (instrument model: Quanta250 of FEI company; magnification: 14–1,000,000 times; accelerating voltage: 200 V–30 kV) to observe the gear slices. The micro-morphology of the wear scar on the gear tooth surface was analyzed. The research provides an important reference value for the research on the wear state of gears of large equipment in the future.

## 2. Materials and Methods

### 2.1. Experimental Preparation

To speed up the test process, shorten the test period, obtain the gear failure data, and observe the obvious tooth-surface-wear effect, the gear sample being tested was thinned using the wire electrical discharge machining (WEDM) method, and the treated tooth width was set to 1/3 of the width of a standard gear tooth. The annealing process was used to reduce the tooth surface hardness of the helical gear. Figure 1 shows the effect of different temperatures on the hardness of 45# steel under the migration of annealing time [19]. The hardness significantly decreases after only 2 min of heat preservation; with the prolongation of the heat preservation time, the hardness continues to decrease, but the change range is small. According to Figure 1, the ideal softening effect is obtained when the heat preservation time is more than 8 min.

In the experiment, 32# Caltex White Oil Pharma was used to accelerate the wear process of the test. The lubricating oil has good oxidation stability, a viscosity index of 32, and a flashpoint of 208 °C. Oil bath lubrication was used in the test gearbox. This method was fully lubricated. The friction surface affected by the oil bath was covered with an oil film during the whole process of wear.

We added 160 mg of Al_2_O_3_ hard particles (270 μm in diameter; mesh: 80) to 800 mL of the test oil sample, and prepared a cup of the oil sample with the same concentration for supplementing the test oil sample after each sampling. Figure 2 shows the impact loading method used in the experiment.

The impact loading was once every 30 min at 40 Nm, and the fixed loading was 25 Nm. The motor speed was set to 1200 r/min, and under the same speed, loading mode, and lubrication conditions, a gear experiment without initial hard particles of Al_2_O_3_ was set for comparison. The gears used in this experiment were 45 steel helical gears, and the gear parameters are shown in Table 1.

### 2.2. Integrated Gear Test Bench

Due to the excellent characteristics of the gearbox, an integrated gear test bench for data acquisition, used for impact load loading, was built. The test bench was divided into a data acquisition module and a working module (Figure 3a). The working module was mainly composed of a driving motor, a test gearbox, a secondary gearbox, an elastic shaft, and a magnetic powder brake. The output torque of the magnetic powder brake was controlled by adjusting the current size of the controller.

The structure diagram of the data-acquisition-integrated test bench is shown in Figure 3b. The data acquisition module consisted of a circulating peristaltic pump, a DH186 piezoelectric acceleration sensor, and a portable data acquisition box. In the experiment, the frequency domain signal of the gear operation was collected. The collected signal was processed in order to judge the gear fault. The circulating peristaltic pump connected the oil filling port at the top of the test gearbox with the oil drain valve at the bottom through a hose, a steel pipe, and a magnetic gauge seat. This connection method was to realize the oil circulation of the gearbox, and meet the requirements of non-stop sampling.

### 2.3. Oil Sample Collection and Preparation

Gearbox oil contains a lot of gear wear information, and it is an important research tool in oil analysis, in order to extract the appearance characteristics, quantitative relationship, and chemical properties of the lubricating oil of abrasive particles [20]. In the test, 20 mL of oil samples were taken from the test gearbox every 60 min. For the obtained oil sample, 5 mL was taken to prepare the spectrum, and the remaining 15 mL was detected by the YJS-170 particle counter, in order to determine the size and quantity of the abrasive particles in the oil. The experiment used a peristaltic pump to ensure the uniformity of abrasive particles in the test gearbox (Figure 4). The prepared ferrograms were observed by a metallographic microscope, to determine the shape of abrasive grains in different wear periods. All instrument specifications and manufacturers in Figure 4 are shown in Table 2.

## 3. Results and Discussion


### 3.1. Wear Particle Monitoring

In the gear wear experiment, due to the circulation of oil in the peristaltic pump, each oil sample obtained reflects the generation of abrasive particles in the entire test gearbox, to a certain extent. Abrasive particles with diameters between 15 μm and 25 μm are used to characterize the change in fatigue wear, and the abrasive particle concentration between 15 μm and 25 μm in the two experiments is plotted (Figure 5).

It can be seen from the figure that the abrasive particle concentration soars to 17,000 particles/mL within 0–20 h in the group of experiments with hard particles at the early stage of wear, due to the influence of hard particles, and then decreases rapidly and remains at 2000 particles/mL for a long time. At this time, it enters the stable wear period. In the experimental group, the abrasive particle concentration rises rapidly to 12,500 particles/mL after 130 h, and then the gear teeth break and fail.

The experiment needs to be shut down every time it is conducted, due to downtime for maintenance. Points 1, 2, and 3 in the figure are the short-term high peaks of abrasive particle concentration every time the machine is restarted. From the graph, we see that the peaks decrease. We infer that there is a “secondary run-in” phenomenon at this time. According to previous studies [21], three-body abrasive wear interrupts the grease film on the gear surfaces, and increases wear. The concentration of wear particles has a similar trend to pitted area growth [22]. From Figure 5, we see that there is a short-term peak in the abrasive particle concentration in the oil after adding the three-body abrasive. This is due to the increased stress concentration on the gear surface in the early stage of wear under the action of the three-body abrasive, which aggravates the abrasive wear. In addition, the aggravation of abrasive wear leads to the enlargement of the pitting area on the gear surface.

### 3.2. Vibration Signal Research

To make the results more obvious, we generally want lower frequency resolution for high-frequency bins, and higher frequency resolution for low-frequency bins. This requires the use of a wavelet transform to process the vibration signal.

In gear fault signal diagnosis, discrete wavelet transformation (DWT) is used for time–frequency analysis of decomposing signals, in both the time and frequency domains. DWT can be defined as [23]:(1)$$DWT(a,b)=\frac{1}{\sqrt{2^{j}}}\int_{-\infty }^{\infty }X(t)\psi^{*}\left(\frac{t-2^{j}k}{2^{j}}\right)dt$$
where a and b represent 2*^j^* and 2*^j^ k* respectively. The DWT analysis of a signal is calculated by passing through a series of filters. The original signal *X(*t*)* can be defined as:(2)$$X(t)=A_{J}+\sum_{j\leq J}D_{j}$$
where *A_J_* and *D_j_* represent the approximation and the detail signals of the *J*th level, respectively. The collected transient signals are used to detect gear faults in the DWT domain.

The frequency spectrum of the vibration signal carries a large amount of gear fault information, which determines whether the gear is abnormally worn [24,25,26]. The experimental speed of the pinion in the wear test in this paper is 1200 r/min. Under this condition, the power frequency f1 of the motor and driving wheel is calculated as 20 Hz, the power frequency f2 of the large gear is 5.122 Hz, and the meshing frequency Fz is 420 Hz. Using squared envelope spectrum analysis, noise reduction is performed through the periodic change of vibration signal energy with time. By constructing analytical signals, the squared envelope spectrum is Fourier-transformed, to extract fault features (Figure 6).

Figure 6a is the vibration signal diagram at the initial stage of wear, at a constant load of 25 Nm. At this time, the gear is in the running-in period, and the load is small. The vibration change mainly comes from the influence of the addition of hard particles. Figure 6b shows the vibration signal diagram under impact loading at the initial stage of wear. It is seen that, compared with Figure 6a, a peak value of 3.276 appears under impact loading, indicating that impact loading has a greater impact on the gear vibration amplitude.

Figure 6c,d are the vibration signal diagram under 25 N m dead load in the later stage of wear and the vibration signal diagram under impact load after gear failure, respectively. The obvious fault signal is seen in the figure, which is due to the abnormal vibration caused by the broken gear and the severe wear of the tooth surface; the fault signal amplitude reaches 10.5 and 11.763.

In the later stage of wear, the surface of the gear is severely worn, and after the experiment is complete, there are many spalling pits on the surface of the gear. Due to the influence of pitting corrosion and fatigue wear on the gear tooth surface, the surface morphology of the gear tooth surface is severely worn, which leads to the increase in gear clearance and aggravation of gear vibration.

### 3.3. Qualitative Analysis of Abrasive Particles

The oil samples are made into spectral slices by analytical demography, and then the wear conditions of the gears at various stages are observed under the metallographic microscope, and several kinds of abrasive particles, commonly found in gear wear, are obtained. At the initial stage of wear, the number of abrasive particles in the experimental group is significantly higher than that in the control experimental group (Figure 7a,b).

There is little difference in the number of abrasive particles during the gear running-in period, and it is seen that the hard abrasive particles have little effect on the stable wear stage of the gear. The abrasive grains in Figure 7c are regular in shape and in the shape of discs and strips, which are generated by the normal wear and tear of the gear friction pair, indicating that the lubrication is good at this time. Figure 7d is the observed chain with large abrasive particles, which appears in the later, severe wear stage of gear wear, where various mixtures are mixed. The sticking matter is presumed to be a mixture of abrasive impurities and a small amount of sludge. The oil sample contains yellow wear debris. It is presumed that the copper alloy parts in the gearbox also experience abnormal wear. This means that, in addition to the serious wear of the gear tooth surface, the bearings and other components also suffer serious wear, and the gear enters a stage of severe wear. Figure 7e,f are the rounded abrasive grains under 300× and 500× mirrors, respectively. It is clear from the figure that the surface of the abrasive grain is smooth, and the boundary is regular.

The presence of excessive wear patterns can cause the lubricating oil system to fail and, at the same time, affect the metallic part [27]. Experiments show that under the action of impact load, the vibration of the friction pair aggravates the rupture of the oil film, resulting in a large number of adhesive abrasive particles and fatigue abrasive particles. Different wear and abrasive particles are produced in different wear stages. If a large number of abnormal abrasive particles are found in the oil, it means that the gear wear is abnormal at this time, which can be used as a means to prevent gear failure [28]. In the later stage of gear wear, a large number of circular and elongated abrasive grain chains appear. This phenomenon proves that the wear condition of the gear can be judged by observing the abnormal abrasive particles in the oil.

### 3.4. Tooth Surface Analysis

As the gear is annealed, its hardness is greatly reduced, and the gear teeth are seriously deformed under the action of impact load before the failure of the gear. The problem is found in the wear amount of the tooth surface, and then the micro-morphological wear is studied. By comparing and observing the failed gear and the intact gear, we see that the degree of wear of the gear tooth root is relatively low, and the tooth above the pitch line wears more seriously, and suffers severe plastic deformation (Figure 8a,b).

To study the change in tooth profile and tooth wear, the Archard equation is introduced to calculate the wear depth of the tooth profile [29]:(3)$$\frac{V}{S}=K\frac{W}{H}$$
where *V* is the wear material volume, *S* is the sliding distance between contact surfaces, *K* is the dimensionless wear coefficient, *W* is the contact load, and H is the surface hardness. The wear depth of any meshing point Q on the tooth profile can be expressed as [30]:(4)$$h_{Q,n}=h_{Q,n-1}+kp_{Q,n-1}s_{Q}$$
where *k* is the friction coefficient, *h_Q,n_* and *h_Q,n_*_−1_ are the nth and (n−1)th wear depths of the Q point, respectively, *S_Q_* is the relative sliding distance p_Q,n_*_−_*_1_ of the Q point, and (n − 1) is the (n − 1) meshing contact stress at time point Q.

Figure 9 is the SEM picture of the tooth surface of the gear slice after wear. Figure 9a shows the strain and deformation under repeated shock loading. This results in a large area of plastic deformation on the tooth surface. Here, the tooth surface forms a network-like plastic deformation structure. Figure 9b shows the surface hardening phenomenon after the plastic deformation of the tooth surface. The material has fallen off at the mark, and the fallen material is deposited in pieces on the bottom of the gearbox. Figure 9c is an SEM image of the material falling off at the tooth root of the gear tooth. During the severe wear period, large pieces of surface material fall off and settle on the bottom of the gearbox. Figure 9d is the SEM picture near the tooth top. At this time, the gear forms two large pits due to the surface material falling off. It is seen that the strength structure of the gear teeth is seriously damaged at this time. Under the action of impact load and alternating stress, the part above the gear pitch line wears more and more seriously, which eventually leads to tooth surface extrusion deformation.

Figure 9e is the SEM image near the index circle on the tooth surface of the control group. Obviously, the wire cutting marks disappear and the surface is smoother, but there are more surface scratches and material peeling marks. Figure 9f is the SEM image near the tooth root of the control group. Since the tooth tip and the tooth root of the other gear are in contact and mesh first, the impact load generated during this process causes more grooves between the tooth tip and the tooth root. Compared with the surface topography of the experimental group gear, it is found that, although the gear tooth surface of the control group is also severely worn, the tooth surface is relatively flat, and there is no serious tooth surface gluing or plastic flow, so the gear is normal at this time. The stability is still good, the gear vibration is relatively small, and the gear has no obvious failure; the gear tooth surface of the experimental group has serious wear forms such as plastic flow, tooth surface gluing, and surface material shedding. The rapid loss of surface material causes damage on the surface of the gear teeth. The flatness is seriously reduced, the strength of the gear is reduced, and it causes strong abnormal vibration of the gear. The abnormal vibration causes the gear to wear more seriously, and finally, the gear fails prematurely.

The initiation of macropitting occurs when fatigue micro-cracks form, due to stress concentrations at the contact surface (related to surface asperities, wear debris, etc.) or sub-surface (related to inclusions, voids, etc.), and the cracks propagate under cyclic loads until the detachment of gear material [31,32]. At the end of the gear wear test, a large area of the surface material near the tooth top falls off, forming a deep pit in which a large amount of oil and abrasive mixture are stored. Experiments show that the wear of the gear below the pitch line is less, and the wear of the gear above the pitch line is more serious.

The contact stress of the tooth surface is an abrupt change. After the stress is repeated many times, several small cracks are generated on the surface near the pitch line near the tooth root. The cracks are filled with lubricating oil. Under the repeated action of the impact, the cracks gradually expand. Finally, this leads to the flake-like peeling of the surface layer to form pitting, as shown in Figure 10a; its enlarged view is shown in Figure 10b.

To further explore the gear wear state, composition and element analysis of the sludge mixture in the pitting pit is carried out. In this experiment, 45 steel is used; its main components are Fe, C, Si, and other elements. After analysis, in addition to these elements, there is also a high content of Gr and Mn elements, and the detection results are shown in Figure 10c below. GCr15SiMn is the material used in the cylindrical roller bearings of the various parts of the gearbox. The existence of these components is because the gears are worn and the bearings are also severely worn, and the generated wear debris enters the lubricating oil to form sludge. This phenomenon verifies the speculation that the bearings, and other components in the gearbox, are also severely worn in the ferrography analysis. It indicates that the tooth-breaking fault occurs in the stage of severe wear of the gear.

## 4. Conclusions

In the gear wear experiment, the real working conditions were simulated by applying impact load to the gearbox and adding three-body abrasive Al_2_O_3_ to the oil. A new comprehensive method was used to study the qualitative relationship between the macro-topography of gears and wear particles. The following conclusions were drawn:The three-body abrasive Al_2_O_3_ accelerates the wear of the initial gear, and the abrasive particle concentration reaches 17,000 per milliliter; the secondary running-in phenomenon occurs during the shutdown maintenance process, and the degree decreases with the number of shutdowns;A group of experiments in which hard particles are added in the later stage of wear generate more abnormal vibrations and a fault signal frequency of 11.763; the impact load at the early stage of wear has a greater impact on gears, and has a greater impact on early wear;The wear scars of the gear tooth surface are observed. Under the repeated action of impact load, the part above the pitch line of the gear tooth surface is severely worn and deformed. Finally, the wear prediction graph of the gear tooth profile is drawn.

## Figures and Tables

**Figure 1 materials-15-04135-f001:**
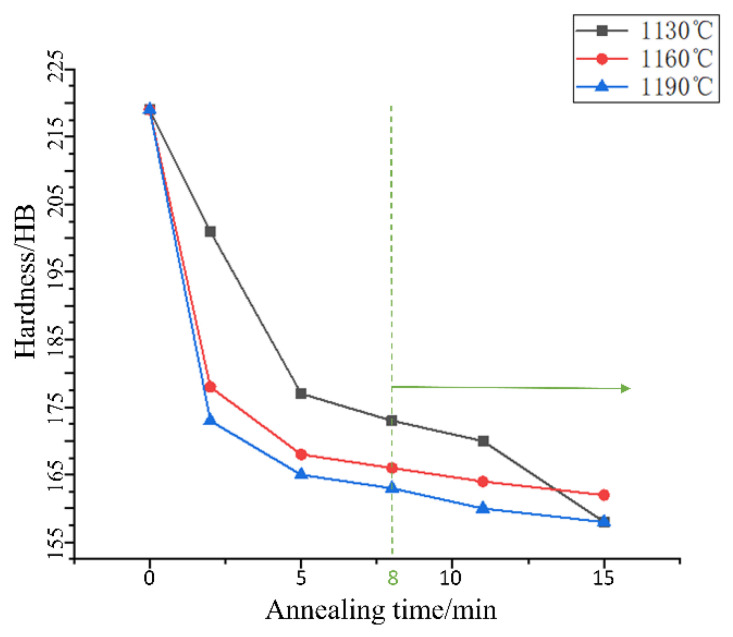
Effect of annealing temperature and time on the hardness of 45# steel.

**Figure 2 materials-15-04135-f002:**
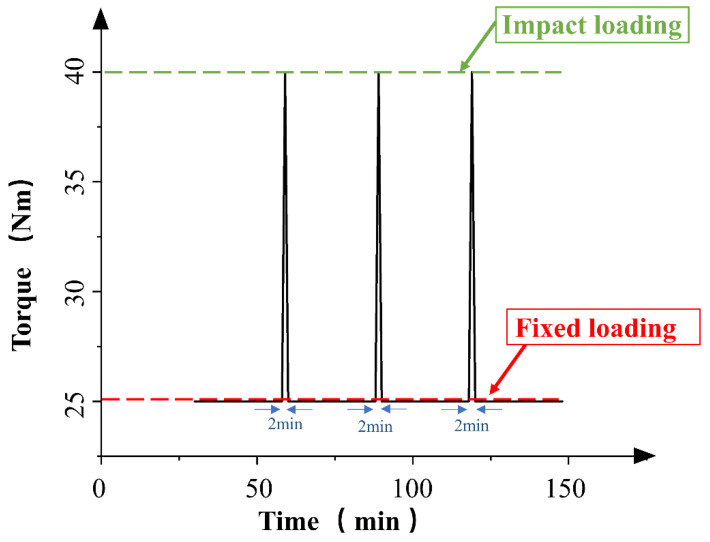
Impact loading mode.

**Figure 3 materials-15-04135-f003:**
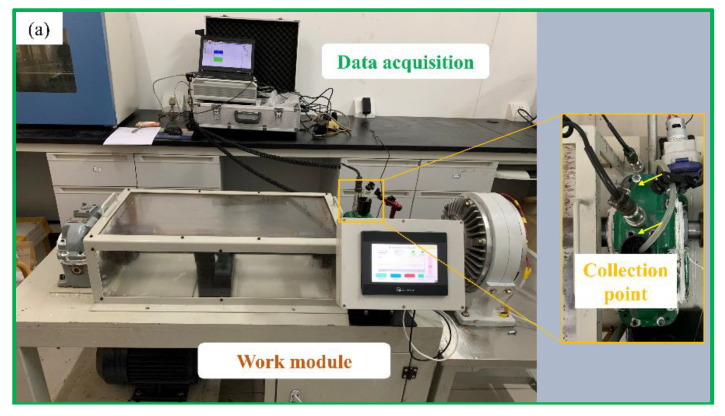
Gear test bench; (**a**) data integration gear test bench, (**b**) composition and structure diagram of test bench.

**Figure 4 materials-15-04135-f004:**
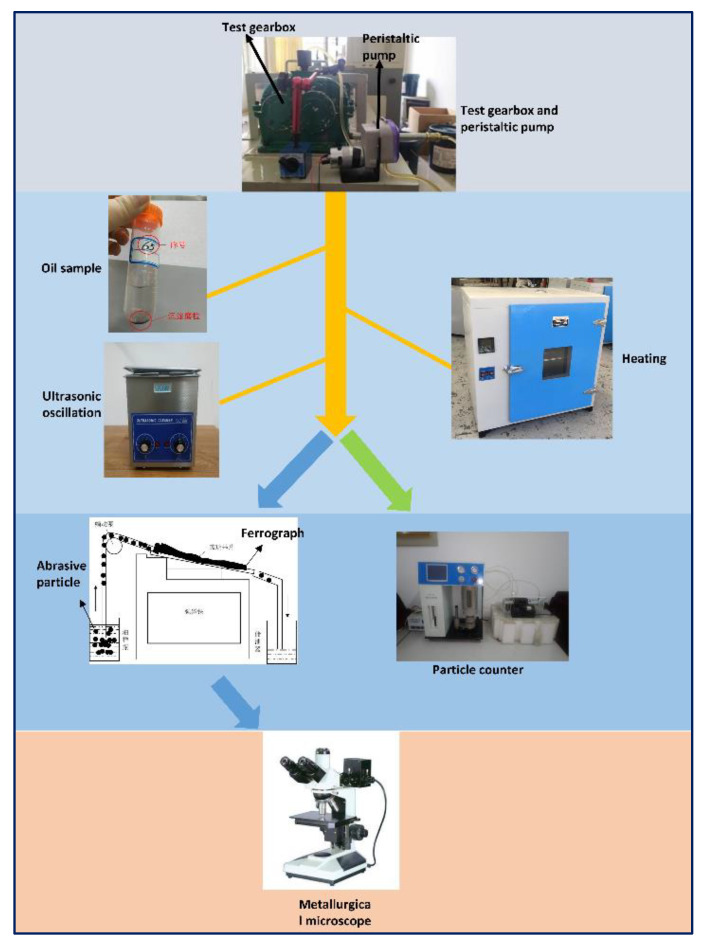
Flow chart of oil sample collection and preparation.

**Figure 5 materials-15-04135-f005:**
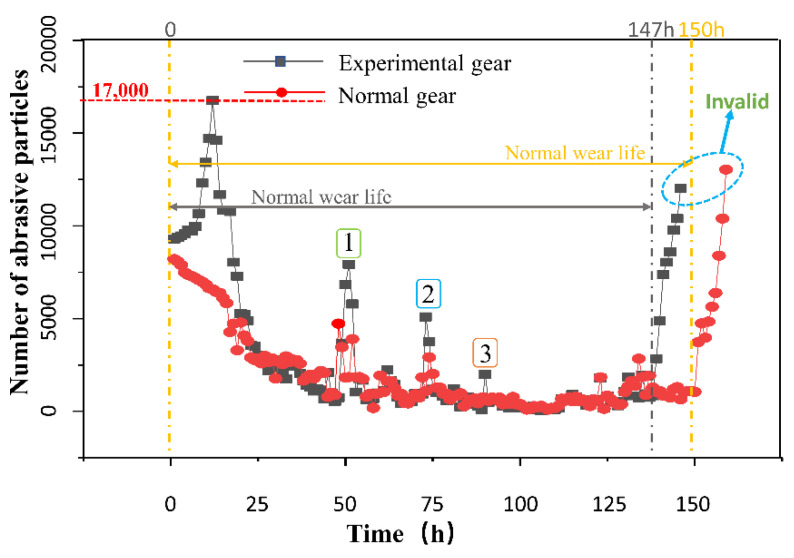
“Bathtub shaped” point line diagram of wear particle change.

**Figure 6 materials-15-04135-f006:**
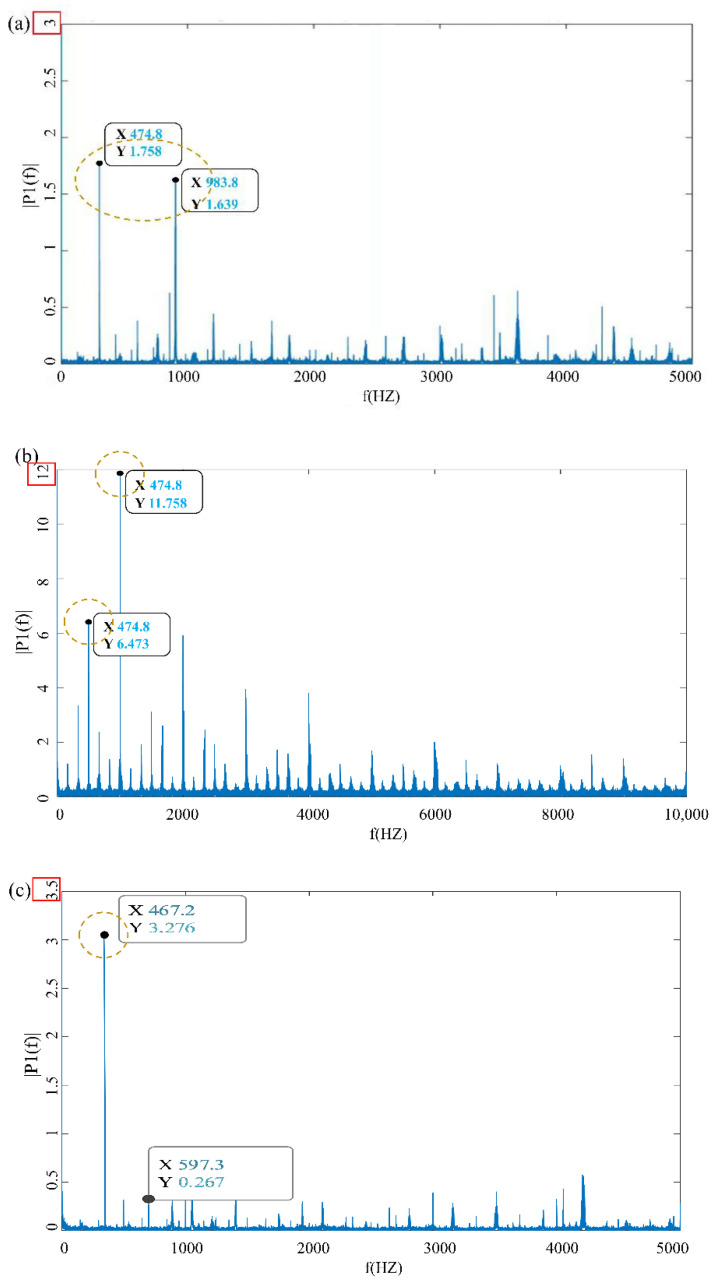
Gear running at each stage spectrum map, spectrums of (**a**) dead load and (**b**) impact loading in running in the wear stage, (**c**) impact loading in the severe wear stage, (**d**) impact loading in failure time.

**Figure 7 materials-15-04135-f007:**
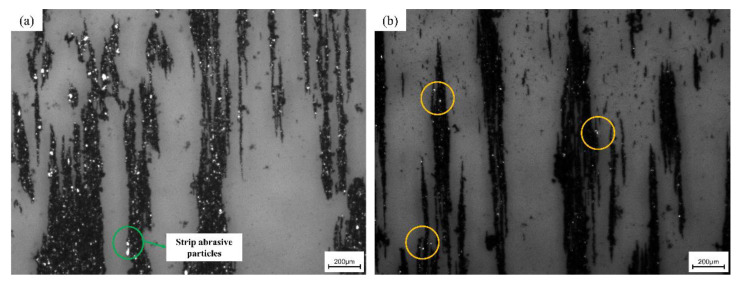
(**a**) Ferrograph of the experimental group in the running-in wear stage; (**b**) ferrograph of the control group in the running-in wear stage; (**c**) 500 times larger abrasive grains in the shape of smooth round and long strips; (**d**) 200 times the large abrasive chain in the severe wear stage; (**e**) smooth, round abrasive grains under 300× magnification; (**f**) Light, rounded large abrasive grains under 500× mirror.

**Figure 8 materials-15-04135-f008:**
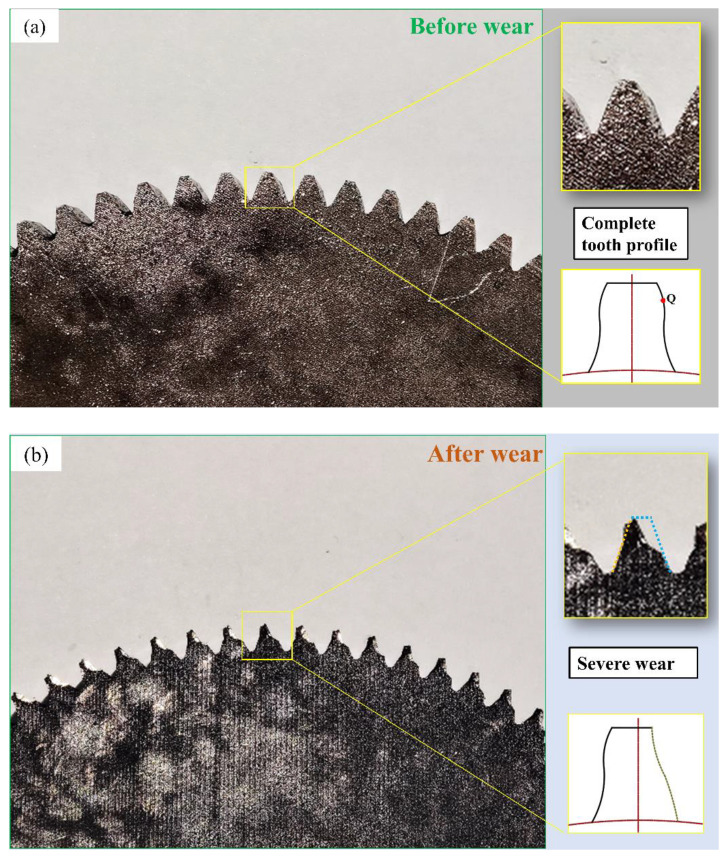
Tooth surface wear; (**a**) gear diagram before the experiment; (**b**) wear failure diagram of experimental gear.

**Figure 9 materials-15-04135-f009:**
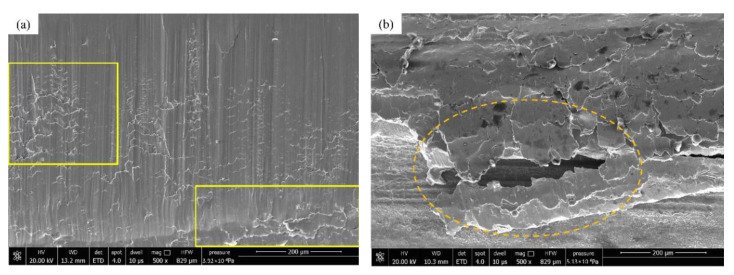
Micro appearance map of the teeth; (**a**) plastic deformation of tooth surface; (**b**) hardening of the tooth surface; (**c**) surface material falling off at the tooth root; (**d**) pit formed by large area of material falling off; (**e**) the groove at the pitch line of the control gear; (**f**) crushing deformation of the gear tooth root in the control group.

**Figure 10 materials-15-04135-f010:**
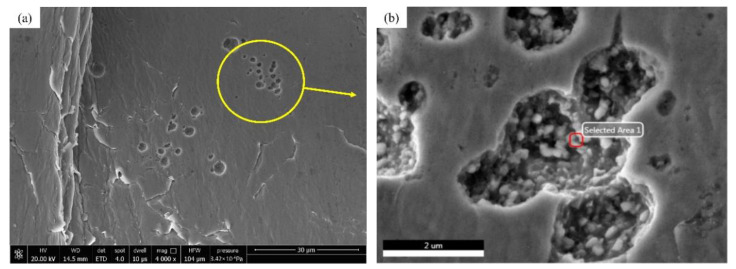
(**a**) Surface pitting diagram; (**b**) enlarged view of pitting pit; (**c**) composition analysis of oil sludge in pitting pit.

**Table 1 materials-15-04135-t001:** Helical gear parameters.

Name	Driving Wheel	Driven Wheel
Number of teeth	21	82
Surface hardness	HB157	HB219
Surface roughness	2~3 μm	
Material science	45#	
Tooth width	10 mm	
Pressure angle	20°	
Modulus	2	

**Table 2 materials-15-04135-t002:** Instrument Specifications and Manufacturers.

	Company Name	Product Number
Gearbox	China Jiangsu Taixing Reducer Co., Ltd.	ZDY80-3.9-IIIJB/T8853-2001
Ultrasonic cleaner	China Dongguan Jiekang Ultrasonic equipment Co., Ltd.	PS-20 A
Blast drying oven	China Shanghai Yiheng Scientific Instrument Co., Ltd.	DHG-9070 A
Ferrograph	China Shenzhen Jiefu Instrument Co., Ltd.	YTF-5
Particle counter	China Shenzhen Yatai Photoelectric Technology Co., Ltd.	YJS-170
Metallographic microscope	China Shanghai Kaixi Technology Co., Ltd.	FL7500

## Data Availability

Not applicable.

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
