# Peer review of "Study on Wear Mechanism of Helical Gear by Three-Body Abrasive Based on Impact Load"

_materials, 2022, doi:10.3390/ma15124135_

Round 1

Reviewer 1 Report

The authors presented an article «Study on wear mechanism of helical gear by three-body abrasive based on impact load».  

·         Abstract

The abstract need to be improved. The abstract is written too long. Shorter and core findings of the study should be given. Please provide the main quantitative and qualitative research core findings. Demonstrate in the abstract novelty, practical significance. Briefly list the input and output parameters of the research.

Al2O3: correct the typo

·         Introduction

In the last paragraph of the introduction section

What is the scientific novelty of the paper? What is the practical value? What makes this approach different from other researchers? Please spesifiy. Gap and significance of the work must be included.

“The experiment was carried out by scanning electron microscope (SEM).” What does this sentence mean? Please fix it.

The introduction of the manuscript can be expanded with articles published in recent years.

·         2. Materials and Methods

There is an error in WEDM. Please fix it. WEDM: wire electrical discharge machining.

“5# steel helical gear” , “ 32# White Oil without” what is?, Please spesifiy

Figure 3 and 4 should be given a closer view. Should be explained in more detail.

Figure 3b figure quality should be increased.

·         3. Results and Discussion

How many times were the experiments repeated?

It is useful to add explanations of parameters to the results obtained. At least five sentences for each Figures. The results obtained should be explained by supporting the literature.

Figure 9 and 10 must be discussed in detail in the article. What are phenomena?

·         Conclusions

The conclusions need to be improved. The results are written too long. It is necessary to more clearly show the novelty of the article and the advantages of the proposed method. Add qualitative and quantitative results of your work. What is the difference from previous work in this area? Show practical relevance. What are the differences from previous works?

·         Authors should carefully study the comments and make improvements to the article step by step. All changes should be highlighted in color.

Reviewer 2 Report

 The manuscript “Study on wear mechanism of helical gear by three-body abrasive based on impact load”, by Wei Yuan, Haotian Wang, Qianjian Guo, Wenhua Wang, Yuqi Zhu, Jie Yu and Xianhai Yang, presents results concerning the wear mechanism evolution, by adding the three-body abrasive Al2O3. Studies were performed due to oil analysis and vibration analysis methods.

The manuscript can be accepted after minor changes. My comments are below.

 1.    Please insert the affiliations for all authors, along with details, including who is the corresponding author.

2.    Insert the subscript for all chemical formula.

3.  Please insert some details regarding SEM microscope (company name, used voltage and magnification, etc).

4.   Please make a short section where you describe all your used instruments (company name, specifications, etc), along with the used software’s. For example, in Figure 4 is shown a microscope and it’s defined as a “Metallurgical microscope”, but from my experience, it’s like a usually lab microscope; and from the image it’s look like a Microscope from Olympus Company.  

5.  Line 81: the authors said:  ...” We added 160 mg of Al2O3 hard particles (270 μm in diameter and 80 meshes) ...”. What does it mean particles with 80 meshes? Because this is understood from the sentence.

6.       Figure 2: on “y” axis is written “ N.m”. What is the purpose of the point between “N” and “m”? you mean a multiplication? If yes, please remove that point. Also, please use the same notation system along the manuscript.

7.       Figure 3: regarding the data acquisition system. What do you acquire? What signal type? Please insert some details.

8.       Figure 5: on “y’ axis is written for the number of abrasive particles the unit “(n)”. What does it mean that “n” unit? Either you delete that “n”, or insert something like “(a.u.)”, from arbitrary unit, otherwise, “n” does not have any signification for numbering something.

9.       Line 129 – 130: authors discuss about particle concentration, e.g., 17000 particles/ ml, but in Figure 5 is shown only the number. Which one is correct? What are you showing in Figure 5? Please make corrections.

10.   Line 133: authors inserted “12500/ ml”. Please use the same notation everywhere. Above, for example in Line 132 appears “2000 particles/ ml”. In this line, please keep the same notation style “value particle/ml”.

11.   Line 150: please define the acronym “DWT”. From where came that acronym?

12.   Figure 7:  In the description of figure 7 is written that are presented grains with spherical-like shapes and long stripes, but the inserted image does not show at all spherical particles. Yes, by seeing something from a higher distance, it seems to be spherical. Please insert some images with a higher magnification, which shows exactly the shape of the grains.
